# Structure of *Salmonella* Flagellar Hook Reveals Intermolecular Domain Interactions for the Universal Joint Function

**DOI:** 10.3390/biom9090462

**Published:** 2019-09-09

**Authors:** Péter Horváth, Takayuki Kato, Tomoko Miyata, Keiichi Namba

**Affiliations:** 1Graduate School of Frontier Biosciences, Osaka University, 1-3 Yamadaoka, Suita, Osaka 565-0871, Japan; 2RIKEN SPring-8 Center and Center for Biosystems Dynamics Research, 1-3 Yamadaoka, Suita, Osaka 565-0871, Japan; 3JEOL YOKOGUSHI Research Alliance Laboratories, Osaka University, 1-3 Yamadaoka, Suita, Osaka 565-0871, Japan

**Keywords:** cryoEM, *Salmonella* hook, universal joint, helical image analysis

## Abstract

The bacterial flagellum is a motility organelle consisting of a rotary motor and a long helical filament as a propeller. The flagellar hook is a flexible universal joint that transmits motor torque to the filament in its various orientations that change dynamically between swimming and tumbling of the cell upon switching the motor rotation for chemotaxis. Although the structures of the hook and hook protein FlgE from different bacterial species have been studied, the structure of *Salmonella* hook, which has been studied most over the years, has not been solved at a high enough resolution to allow building an atomic model of entire FlgE for understanding the mechanisms of self-assembly, stability and the universal joint function. Here we report the structure of *Salmonella* polyhook at 4.1 Å resolution by electron cryomicroscopy and helical image analysis. The density map clearly revealed folding of the entire FlgE chain forming the three domains D0, D1 and D2 and allowed us to build an atomic model. The model includes domain Dc with a long β-hairpin structure that connects domains D0 and D1 and contributes to the structural stability of the hook while allowing the flexible bending of the hook as a molecular universal joint.

## 1. Introduction

The bacterial flagellum is a motility organelle and is a complex nanomachine consisting of more than 20 different proteins in different copy numbers ranging from a few to a few tens of thousands [1]. Bacteria swim in viscous liquid toward more favorable environments from less favorable ones for their survival, proliferation and/or infection [2,3,4]. The bacterial flagellum can be divided into three main parts: the transmembrane basal body that acts as a rotary motor as well as the flagellar protein export machine to construct the flagellar axial structures; the long filament extending into the cell exterior to function as a helical propeller; and the hook connecting the motor and filament as a universal joint to transmit motor torque to the filament. 

The rotary motor of *Salmonella* is powered by proton motive force across the cell membrane and rotates the filament at around 300 revolutions per second [5,6,7]. The filament is a helical assembly of a single protein, FliC (flagellin), and a few tens of thousands of FliC molecules form a 10–15 µm long supercoiled tubular structure in a gently curved form to be a helical propeller [1], which is rotated by the motor to produce thrust for bacterial swimming [8,9]. The filament is normally a left-handed supercoil, and its counter-clockwise rotation by the motor produces thrust for cell swimming. *Salmonella* has several peritrichous flagella that form a bundle behind the cell to produce strong thrust, but the switching of motor rotation to clockwise direction causes a polymorphic transition of the filament to a few different right-handed supercoils, thereby making the bundle to fall apart, causing cell tumbling for chemotaxis and thermotaxis [10,11,12]. The relatively rigid structure of the filament against bending assures these left- and right-handed supercoils function as a propeller. 

The *Salmonella* hook is a short, highly curved tubular structure built by helical assembly of about 120 copies of a single protein, FlgE [13,14,15,16]. The length of the hook is regulated to 55 nm ± 7 nm [17], and its highly curved structure and bending flexibility make it work as a universal joint, transmitting motor torque to the filament regardless of the filament orientation off-axis of the motor during run and tumble of the cell for taxis [18,19,20]. 

The distal end of the hook is connected to the filament via two hook-associated proteins, FlgK and FlgL, while its proximal end is directly connected with the rod within the basal body. The rod is rigid as a drive shaft that transmits motor torque to the filament through the hook. The rod, hook and filament form the axial structure of the flagellum and show structural similarity to each other as tubular structures composed of 11 protofilaments. Nevertheless, their diameters, mechanical properties and functions are distinct from each other. The rod consists of four different proteins, FlgB, FlgC, FlgF, and FlgG [21] in different copy numbers: around five for FlgB, FlgC and FlgF; and about 26 for FlgG [22]. The distal, major part of the rod is formed by FlgG, which shows a high (32%) sequence similarity to FlgE, and the structure of the polyrod formed by a FlgG mutant shares the same helical symmetry and axial repeat distance with that of the hook, explaining why the hook is directly connected to the rod without any adaptor proteins between them [23]. 

The structure of the bacterial flagellum has been intensively studied for *Salmonella enterica* serovar Typhimurium by X-ray crystallography and electron microscopy (cryoEM) for many different parts [24]. The crystal structure of a core fragment of hook protein FlgE as well as the structure of polyhook, which is an abnormally long hook produced by deletion of the hook ruler protein FliK, are also available for the *Salmonella* flagellum [25,26,27], and these structures provided deep insights into the structural mechanism for the universal joint function. The axial packing interactions of the FlgE subunits along the protofilament of the hook have evolved in such a way for the protofilament to be compressible and extensible by mechanical force to make the tubular structure made of 11 protofilaments quite flexible in bending while their lateral packing interactions keep the tubular structure stable to make it rigid against twisting. The FlgE molecule consists of three major domains, D0, D1 and D2, that are arranged radially from the inner core to the outer surface of the tubular structure and axially from the proximal to the distal end of the hook. The structures of these domains are well resolved by combined use of X-ray crystallography and cryoEM helical image analysis [25,26,27]. The FlgE chain is folded to form these three domains by starting from D0, going through D1 and D2, and coming back to D0 through D1. In addition, a previous cryoEM study identified a small, elongated domain Dc that connects domains D0 and D1, and this domain appears to play an important role in stabilizing the hook structure by intervening the packing interactions of neighboring subunits, but it was difficult to trace the chain to build an atomic model due to the resolution of the map being limited to 7.1 Å (at a Fourier shell correlation of 0.5), even though most of the secondary structures in the other three domains were clearly visible [23]. A recent high-resolution cryoEM study on the hook structure of *Campylobacter jejuni* revealed the domain structures that are nearly identical to those of *Salmonella* hook FlgE and also resolved the structure of domain Dc clearly as an L-shaped, long β-hairpin named L-stretch [28]. This long β-hairpin structure appears to fit well into the previous density map of domain Dc of *Salmonella* hook, but the length of the *Salmonella* peptide chain assigned to this Dc domain is much shorter than that of *Campylobacter* by 17 residues. So a question still remains as to how domain Dc with the β-hairpin conformation interacts with neighboring FlgE subunits to stabilize the entire hook structure without reducing its bending flexibility. 

We therefore used cryoEM image analysis to obtain a high-resolution structure of *Salmonella* hook to build its complete atomic model. With recent advances in cryoEM techniques with the use of a direct electron detector camera for high-resolution image data collection, we have obtained a 3D density map at 4.1 Å resolution and built an atomic model that reveals the conformation of domain Dc and its intimate interactions with neighboring subunits. We describe the nature of these intersubunit interactions and discuss the role of domain Dc in the structural stability and universal joint function of the hook.

## 2. Materials and Methods 

### 2.1. Sample Preparation and Electron Microscopy 

Flagellar polyhooks were isolated from the SJW880 strain of *Salmonella enterica* serovar Typhimurium (Δ*fliK* mutant) according to the protocol by Aizawa et al. with slight modifications [29]. After isolation and purification, the polyhook solution was kept at 4 °C in a cold room overnight to make polyhooks straight. A 3.0-µL sample solution was applied onto a Quantifoil holey carbon molybdenum grid (R0.6/1.0, Quantifoil Micro Tools GmbH, Jena, Germany), and the grid was plunge-frozen into liquid ethane using Vitrobot (Thermo Fisher Scientific, Waltham, MA, USA). The grid was imaged manually with a JEM-3200FSC electron cryomicroscope (JEOL, Tokyo, Japan), equipped with a field emission electron gun operated at 300 kV, an Ω-type in-column energy filter with the energy-selection slit width set to 10 eV for zero-loss imaging, and a K2 Summit Direct Electron Detection Camera (Gatan, Pleasanton, CA, USA). The grid temperature was kept at 77 K by liquid N_2_, and cryoEM images were recorded in the Super-Resolution mode of K2 camera with an image size of 7676 × 7420 pixels, at a nominal magnification of 50,000×, a defocus range of 0.5–2.5 µm, and a dose rate of 2.2 electrons/pixel/sec. Images were recorded in a dose fractionation mode, with a total exposure time of 6 s at a frame rate of 0.2 s/frame, and so 30 frames were recorded for each movie image, with a total cumulative dose of 82 electrons/Å^2^.

### 2.2. Image Processing

Beam-induced sample motions on dose-fractionated frames were corrected by GPU accelerated MotionCor2 software [30]. First global motion correction was carried out on individual frames, and subsequently the corrected frames were divided into 5 × 5 patches and local motion was corrected. Gain reference and dose weighting was then applied, the first three frames were discarded, and the rest were averaged. Accurate determination, refinement and correction of the contrast transfer function was carried out by Gctf, a GPU-accelerated software [31]. RELION 2.0 [32] was then used for image processing and 3D image reconstruction. 

In total, 42,293 segment images were extracted from 486 micrographs, 2D classification was performed, and 41,260 segment images were used for further image processing. In the 3D classification step, local search of helical symmetry (the twist angle and axial rise per unit) was performed with initial parameters of 64.7° for the twist angle and 4.1 Å for the axial rise [23]. In the 3D reconstruction, auto-refine step local search was repeated with initial parameters of 64.78° for the twist angle and 4.05 Å for the axial rise and a step size of 0.1 Å. The helical parameters converged nearly to the initial parameters, confirming the accuracy of the helical parameters. In the mask creation step prior to binarization, the input 3D map was lowpass-filtered to 10 Å resolution. The initial binary mask was extended with a raised-cosine soft edge with 15-pixel width in every direction. In the post-processing step, map sharpening was performed with the given detector MTF and a B-factor of −216. Resolution was determined by calculating the Fourier shell correlation (FSC) coefficient between two halves of the data by the gold-standard approach to avoid overfitting. The final resolution was 4.06 Å at the FSC of 0.143 (Appendix A).

### 2.3. Atomic Model Building

The previous atomic model of *Salmonella* hook (PDB ID: 3A69) [27] with an FlgE subunit consisting of domain D0, D1 and D2 was used as the initial model to fit into the 3D map domain by domain. The atomic model of domain Dc was built by a homology model-building software, MODELLER [33]. The target sequence was that of *Salmonella* FlgE domain Dc, and the 3D model template was the corresponding domain Dc of *C. jejuni* FlgE (PDB ID: 5JXL) [28]. The sequence identity between these two Dc domains was 35%. Then this homology model was fit into the 3D map by Rosetta [34] and refined with its asymmetric refinement procedure. The Ramachandran plot by Coot software [35] showed that 94% of the residues were in the preferred regions and 5% were in the allowed regions. 

A homology model of *Salmonella* FlgG including domain Dc was also built by MODELLER [33] using a recent cryoEM model (PDB ID:5WRH) [23] as a template. The target sequence was that of *Salmonella* FlgG, and the 3D model template was again the corresponding Dc domain of *C. jejuni* FlgE (PDB ID: 5JXL) [28] as well as domains D0 and D1 of *Salmonella* FlgG (PDB ID:5WRH) [23]. The Ramachandran validation in Coot software showed that 89% of the residues were in the preferred regions and 7% in the allowed regions. The relaxation of the structure by searching the local conformation space was carried out by Rosetta Relax [34] by a modified scope of the Relax program, restricting the conformations that Relax can sample. All atomic model and maps are rendered by UCSF Chimera [36]. 

The cryoEM 3D density map has been deposited to the Electron Microscopy Data Bank (https://www.ebi.ac.uk/pdbe/emdb/) under accession code EMD-9974, and the atomic model coordinates have been deposited to the Protein Data Bank (https://pdbj.org) under accession code 6KFK.

## 3. Results and Discussion

### 3.1. Structure Determination

The hook was purified from a *fliK*-deficient mutant strain of *Salmonella*, SJW880 [37], which produces polyhooks that are structurally identical to the native hook but can grow as long as 1 µm. The polyhook solution was incubated at 4 °C overnight to convert the supercoiled form into being straight, a 3 µL of the solution was applied onto a Quantifoil holey carbon grid, and then the grid was plunge-frozen into liquid ethane using a vitrification device (Vitrobot, TFS) [23]. The grid was then observed by a JEOL electron cryomicroscope, JEM-3200FSC, equipped with an Ω-type energy filter, a Schottky-type field emission electron gun operated at 300 kV, and a Gatan K2 Summit direct electron detector camera in movie mode. Movie frames were corrected with MotionCor2 software [30]. Image selection criteria were based on the Thon ring evaluation, and image analysis was carried out by RELION 2 [32]. The final resolution of the three-dimensional image reconstruction was 4.1 Å by the criteria of a Fourier shell correlation of 0.143. The density map was used to build an atomic model, starting from a previous model of domains D0, D1 and D2 of *Salmonella* hook [23] and a homology model of domain Dc generated by MODELLER software [33] based on the model of *Campylobacter* hook [28]. The atomic model was refined through iterative local rebuilding by ROSETTA software [34]. 

### 3.2. Structure of Salmonella Hook and FlgE in the Hook

The 3D density map and the model of *Salmonella* hook are presented in Figure 1 in different views and slices to visualize the structure and helical array of each of the four domains in different colors: domain D2 in magenta; D1 in blue; Dc in green; and D0 in red, from left to right panels. In the previous model built based on a cryoEM density map [23], the model of domain D0 was not so accurate and that of domain Dc was missing due to limited resolution of the density map. The model of *Salmonella* FlgE is now complete with the full-length chain connected from the N- to the C-terminus as shown in Figure 2 in two different views. Domain D0 consists of the N- and C-terminal α-helices (Ser 1–Ala 27 and Leu 367–Arg 402, respectively) forming an antiparallel coiled coil, with the C-terminal helix facing the central channel and tilted from the hook axis by about 17°. Domain Dc is a long β-hairpin (Lys 32–Phe 60) running parallel with the N-terminal helix and with its length nearly the same as that of the N-terminal helix. Thus, domains D0 and Dc together form a compact domain, which we hereafter call domain D0-Dc. A stretched chain after the β-hairpin of domain Dc (Thr 61–Gly 71) and another stretched chain before the C-terminal helix (Gly 358–Asp 366), together with a loop connecting the N-terminal helix and β-hairpin (Thr 28–Phe 31), forms the surface of domain D0-Dc closely interacting with domain D1. Domains D1 (Leu 72–Lys 146 and Tyr 283–Asn 357) and D2 (Ser 147–Gly 282) both form compact globular domains consisting of β-sheets, β-hairpins and loops, as was revealed by X-ray crystallography of a core fragment of FlgE [25]. The pair of chains connecting domains D0-Dc and D1 and those connecting domains D1 and D2 both appear to be flexible enough for each of the 11 stranded hook protofilaments to be curved and twisted to different extents and directions when the hook is bent to different curvatures for the universal joint function.

### 3.3. Intersubunit Interactions

Intersubunit interactions are shown for each of the four domains in the upper panel of Figure 1. FlgE subunits assemble into the tubular structure of the hook in a helical array, and the hook structure has three major helical lines: −5-start, 6-start and 11-start, where the number represents the number of helical strands, and the negative and positive numbers mean left- and right-handed helices, respectively. According to these helical lines, we call one of the subunits subunit 0 and its surrounding six subunits as subunit −5, 5, −6, 6, −11, and 11 (see upper panel of Figure 1). 

The D2 domains (magenta) form continuous helical arrays along the 6-start helical lines on the hook surface but have no interactions along −5-start and 11-start helices, and therefore, these six stranded helical arrays of D2 domains are widely apart from each other. The D1 domains (blue) form a mesh-like array along all three major helical lines but their interactions are not very tight with a small gap in all the directions, suggesting that these interactions modestly contribute to the stability of the hook structure. The D1 domains are probably interconnected indirectly through one or two layers of water molecules between their domain surfaces, although water molecules are not visible at this resolution [38]. These large and small axial gaps between the D2 domains and between the D1 domains, respectively, are actually important for flexible bending of the hook for its universal joint function because the axial compression and extension of the protofilaments are essential for the bending motion.

In contrast, the Dc domains (green) have close interactions in the −5-start and 11-start helical lines, making a stable tubular structure, albeit the 6-start interactions are absent. The D0 domains (red) form intimate interactions in all the directions, with the N- and C-terminal helices forming a two-layer helical bundle where the C-terminal helices tile the inner surface of the central channel of the hook with polar and charged residues. The N-terminal helices, not facing the channel, have interactions only in the −5-start helical line, but together with Dc domains, they are involved in intricate intersubunit interactions. As described in the previous section, the N-terminal helix of domain D0 and the long β-hairpin of domain Dc form a tight interaction, strongly suggesting that domains D0 and Dc are likely to behave as one domain D0-Dc. A stretched loop of subunit −11 formed by residues 64–67 at the top of domain Dc connecting to domain D1 is involved in close tripartite interactions with the tip of the long β-hairpin of domain Dc (residues 40–52) and the N-terminal end (residues 1–7) of subunit 0 (Figure 3 upper panel). Thus, the tubular structure of the hook is stabilized and maintained mainly by the intersubunit interactions of D0-Dc domains alone in the inner core of the hook structure. 

There is, however, an additional intersubunit interaction contributing to the structural stability that is found between different domains. Residues 38–40 in the middle of the β-hairpin of domain Dc of subunit 0 are closely interacting with the tip loop (residues 328–331) of a short β-hairpin of domain D1 of subunit −5 (Figure 3 lower panel). Thus, domain D1 is also involved in further stabilizing the hook structure through an intersubunit interaction with domain D0-Dc. Thus, these intersubunit interactions clearly indicate the importance of the long β-hairpin of domain Dc for the structural stability of the hook, as found by deletion mutation experiments [39].

Even in these tight interactions between D0-Dc domains stabilizing the hook structure, the axial gap with a distance corresponding to about one turn of α-helix (5 Å) is still present to allow the compression of the protofilament for flexible bending of the hook (Figure 3 upper middle panel). The mutual displacement of the chains involved in these 11-start tripartite interactions may be small enough to be achieved by conformational rearrangements of their side chains but an intersubunit sliding interaction similar to what occurs between the triangular loop of domain D1 and domain D2 of subunit −11 [25] may also occur upon bending of the hook.

### 3.4. Role of the Longer β-Hairpin of Domain Dc in the Flagellar Rod Structure

The flagellar rod is a drive shaft of the rotary motor, transmitting motor torque to the filament acting as a helical propeller through the hook as a universal joint. The rod is a helical tubular assembly of four proteins, FlgB, FlgC, FlgF, and FlgG, but its major part rotating inside the LP ring of the flagellar basal body is the distal part of the rod formed by FlgG [23]. The diameter of the FlgG rod is only 13 nm, markedly smaller than 18 nm of the hook, but the rod is much more rigid as a drive shaft than the hook working as a universal joint. Interestingly, however, *Salmonella* FlgG and FlgE share a highly homologous sequence with 32% identity by BLAST search [40], except that the FlgG sequence is shorter than FlgE by the absence of 141 residues (residues 144–284) in the central region of the FlgE sequence corresponding to domain D2, and so FlgG consists of domains D0-Dc and D1 and is missing D2. That is why the diameter of the rod is smaller than that of the hook. 

On the other hand, FlgG has a region of 18 extra residues in the domain Dc region, suggesting that the β-hairpin of domain Dc is longer than that of FlgE (Figure 4a). FlgE of *C. jejuni* also has a similar sequence to *Salmonella* FlgG, with 17 extra residues in this region compared with *Salmonella* FlgE (Figure 4b), suggesting a longer β-hairpin in *Campylobacter* FlgE than *Salmonella* FlgE as well. In agreement with this prediction, the recent cryoEM structure of the hook isolated from *C. jejuni* showed an L-shaped, long β-hairpin named L-stretch in domain Dc of FlgE [28], which is much longer than that of *Salmonella* FlgE. So we used this structure to build a homology model of the β-hairpin of *Salmonella* FlgG and inserted it into the atomic model of FlgG in the polyrod solved by cryoEM analysis [23] in order to compare the differences in these three structures. 

Pairwise comparisons by superposing the D0 domains are shown in Figure 5 with *Salmonella* FlgE in blue, *Campylobacter* FlgE in red and *Salmonella* FlgG in green. The relative disposition of the β-hairpin to the N- and C-terminal helices of domain D0 is nearly the same between *Salmonella* FlgE and *Campylobacter* FlgE (Figure 5 left) and is slightly different between *Salmonella* FlgG and *Campylobacter* FlgE (Figure 5 middle) and between *Salmonella* FlgG and FlgE (Figure 5 right), but these differences are mainly in the distal portion of the β-hairpin. The model of the FlgG rod, which was missing domain Dc in the previous study [23] but is now completed with the long β-hairpin, is shown in Figure 6. The model is shown in different views and slices similarly to Figure 1 to visualize the structure and helical array of each of the three domains in different colors: D1 in yellow; Dc in magenta; and D0 in light blue, from left to right. The distinct structural feature of the rod compared to that of the hook shown in Figure 1 is that the tip of the β-hairpin of domain Dc is projecting out (Figure 6 middle). If we look at the β-hairpin structures and their locations and interactions with other domains in *Salmonella* hook, *Salmonella* rod and *Campylobacter* hook, as presented in Figure 7, it is clear that the long β-hairpins of *Salmonella* FlgG and *Campylobacter* FlgE are projecting out into the outer layer of their tubular structures composed of the D1 domains and are filling the gap formed in the helical array of the D1 domains to further stabilize and rigidify the tubular structure. 

Although *Campylobacter* hook is curved in the native form and functions as a universal joint, it easily becomes straight and appears to be more rigid than *Salmonella* hook [28], and this is well explained by the tip of long β-hairpin filling the gap of the helical array of the D1 domains. *Campylobacter* hook needs to be more rigid than *Salmonella* hook, which is maybe because *Campylobacter* has two polar flagella and the hook needs to be rigid in order to keep orienting the flagellar filaments in the axis of the cell body to produce a thrust efficiently. The insertion of extra 18 residues of the *Salmonella* FlgG-specific sequence into *Salmonella* FlgE was expected to make *Salmonella* hook as rigid as *Salmonella* rod. The hook formed by this insertion mutant of FlgE actually became straight and rigid just like the rod, and the flagellar filaments on these mutant *Salmonella* cells were all spread apart and could not form a bundle to produce thrust for swimming even though individual motors were rotating rapidly [18]. 

Thus, the long β-hairpin of domain Dc plays an important role in the flagellar rod and hook structure, regulating and optimizing the structural stability and rigidity of the entire tubular assembly by changing its length for their required functions. The cryoEM structures of the flagellar rod and hook clearly revealed it, demonstrating the power of the cryoEM technique as a unique and essential tool for structural biology capable of visualizing native, functional structures of biological macromolecular assemblies. 

## 4. Conclusions

We carried out the structural analysis of the flagellar hook isolated from *Salmonella* in the straight form by cryoEM helical image analysis and built an atomic model based on the 4.1 Å resolution map. This model revealed the entire structure of the hook component protein FlgE consisting of domains D0, D1 and D2 as well as the long β-hairpin connecting domains D0 and D1, which is involved in intricate intermolecular interactions to stabilize the tubular structure of the hook while maintaining its bending flexibility. This structural feature is distinct from that of the flagellar rod consisting of FlgG with a much longer β-hairpin, which further stabilizes and rigidifies the rod structure to work as the drive shaft of the flagellar motor. 

## Figures and Tables

**Figure 1 biomolecules-09-00462-f001:**
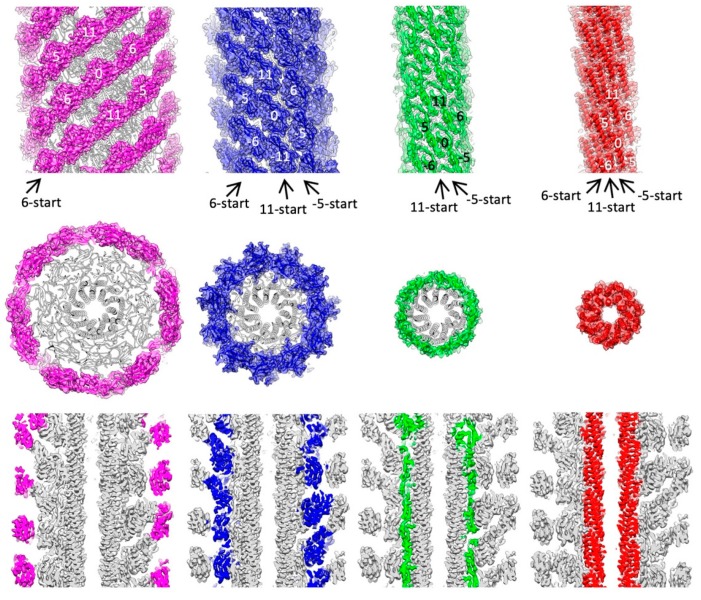
Structures and spatial arrangements of FlgE domains in the hook. The 3D density map and ribbon diagram of the atomic model are presented in different views and slices to visualize the structure and helical array of each of the four domains in different colors: D2, magenta; D1, blue; Dc, green; and D0, red, from left to right panels. The side views are shown in the upper panels, end-on views in the middle, and central slices along the hook axis in the lower. The diameters of these domain arrays are 172 Å, 142 Å, 85 Å, and 62 Å, respectively. The start numbers of the three major helical lines are labeled with arrows in the side views.

**Figure 2 biomolecules-09-00462-f002:**
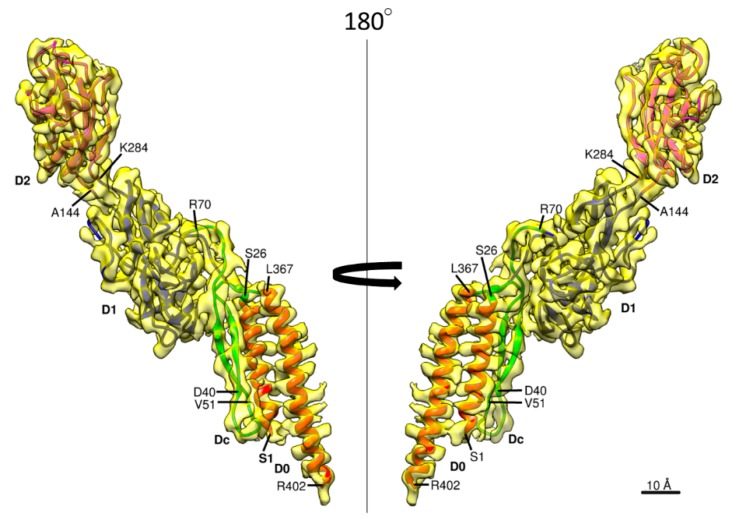
The 3D density map and model of a FlgE subunit. Two side views are shown, with the four domains colored as follows: D0, red; Dc, green; D1, gray; and D2, orange.

**Figure 3 biomolecules-09-00462-f003:**
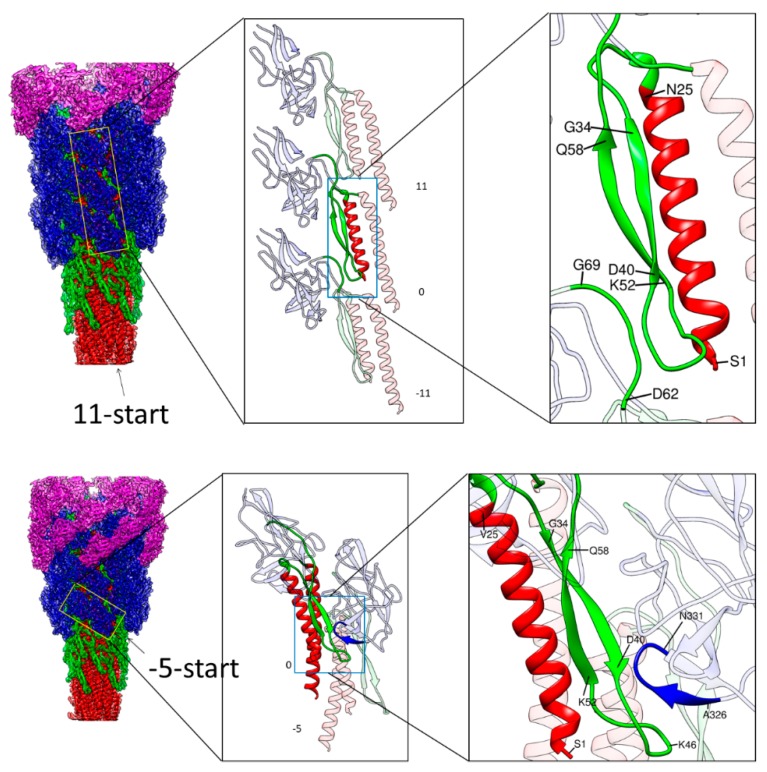
Intermolecular interactions along the 11-start and −5-start direction. Left panels are side views of the hook with the four domains of FlgE colored as in Figure 1, with outer domains removed towards the bottom. The yellow boxes in the left panel indicate the arrays of FlgE subunits magnified in the middle panel: Three subunits in the 11-start helical line in the upper row and two subunits in the −5-start in the lower row. Magnified images of some parts of FlgE are also colored as in Figure 1 in the middle and right panels. In the upper row, the N-terminal helix and the long β-hairpin of domain Dc of subunit 0 are colored red and green, respectively. In the lower row, the N- and C-terminal helices and the long β-hairpin of subunit 0 are colored red and green, respectively, and the tip loop of a short β-hairpin at the bottom of domain D1 of subunit −5 is colored blue. These colored parts in the middle panels are highlighted with light-blue boxes to indicate the portions further magnified in the right panels.

**Figure 4 biomolecules-09-00462-f004:**
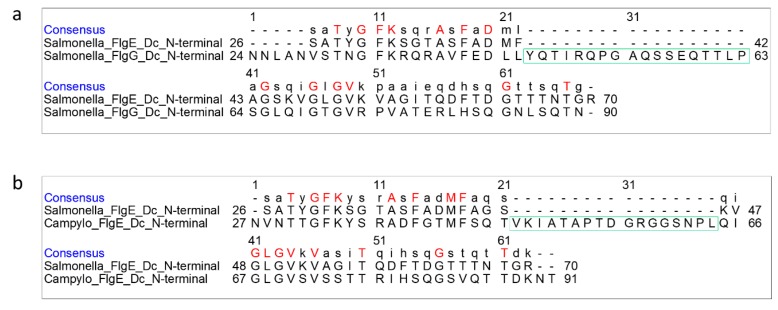
Pairwise sequence alignment of FlgE and FlgG in the domain Dc region. (**a**) Sequence alignment between *Salmonella* FlgE and FlgG, and (**b**) between *Salmonella* FlgE and *Campylobacter* FlgE. The 18 and 17 residue insertions in *Salmonella* FlgG and *Campylobacter* FlgE, respectively, are boxed in green.

**Figure 5 biomolecules-09-00462-f005:**
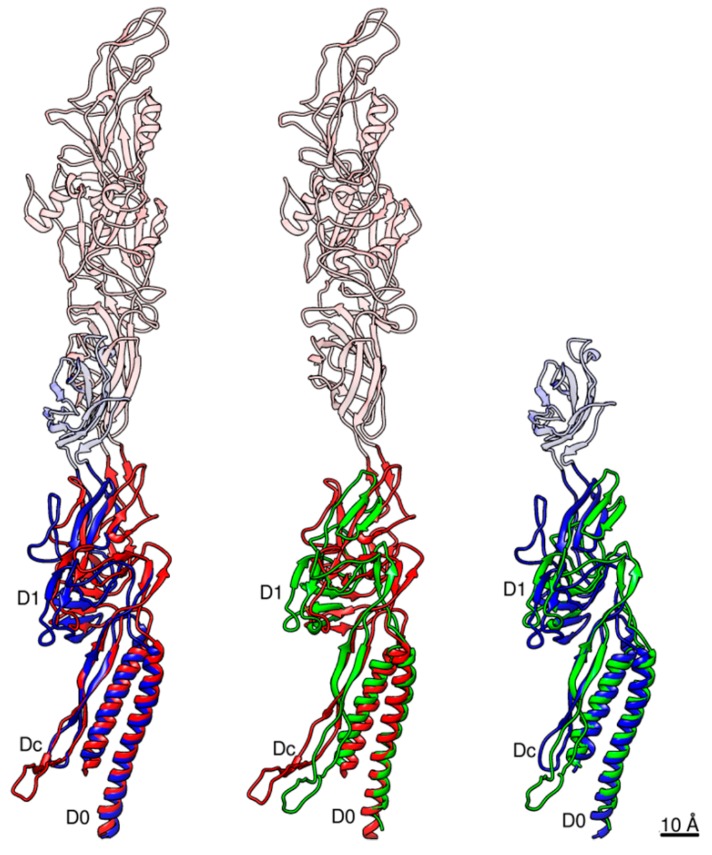
Pairwise structural comparison between *Salmonella* FlgE and FlgG and *Campylobacter* FlgE. *Salmonella* FlgE and *Campylobacter* FlgE are compared in the left panel, *Salmonella* FlgG and *Campylobacter* FlgE in the middle, and *Salmonella* FlgE and FlgG in the right. The Cα ribbon models are colored only for domains D0-Dc and D1, with *Salmonella* FlgE in blue, FlgG in green and *Campylobacter* FlgE in red. Domain D0 is used to superpose these molecules. The relative arrangements of domains D1 and D0-Dc between these three molecules are slightly different to one another as shown in this figure. The extra part of domain D1 present in FlgE and missing in FlgG is the triangular loop involved in the intersubunit sliding interactions between D1 and D2 along the protofilament for its compression and extension.

**Figure 6 biomolecules-09-00462-f006:**
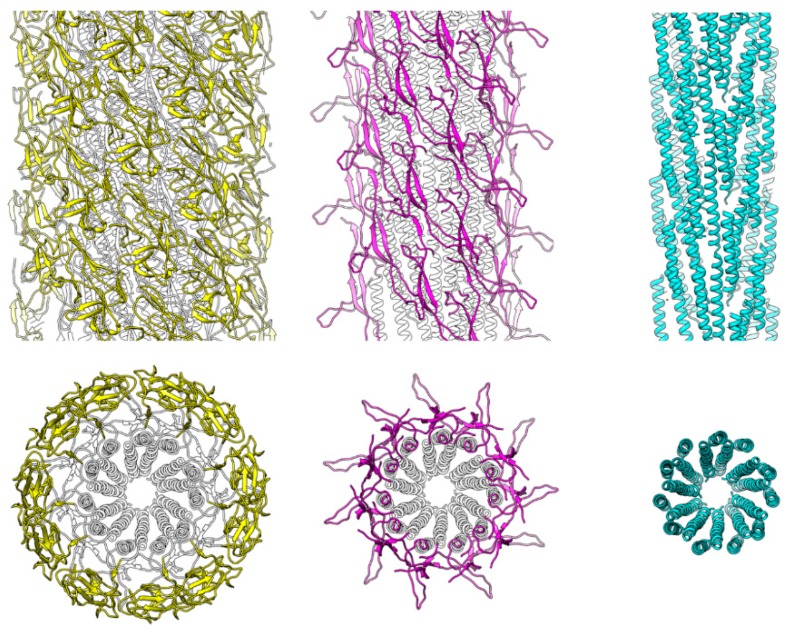
Structures and spatial arrangements of FlgG domains in the rod. The Cα ribbon models are presented in side and end-on views in the upper and lower panels, respectively, to visualize the structure and helical array of each of the three domains in different colors: D1, yellow; Dc, magenta; and D0, light blue, from left to right. The structure of FlgG is very similar to that of FlgE except for the length of the β-hairpin of domain Dc, with the extra length formed by the FlgG-specific 18 residue insertion projecting out nearly horizontally in the rod.

**Figure 7 biomolecules-09-00462-f007:**
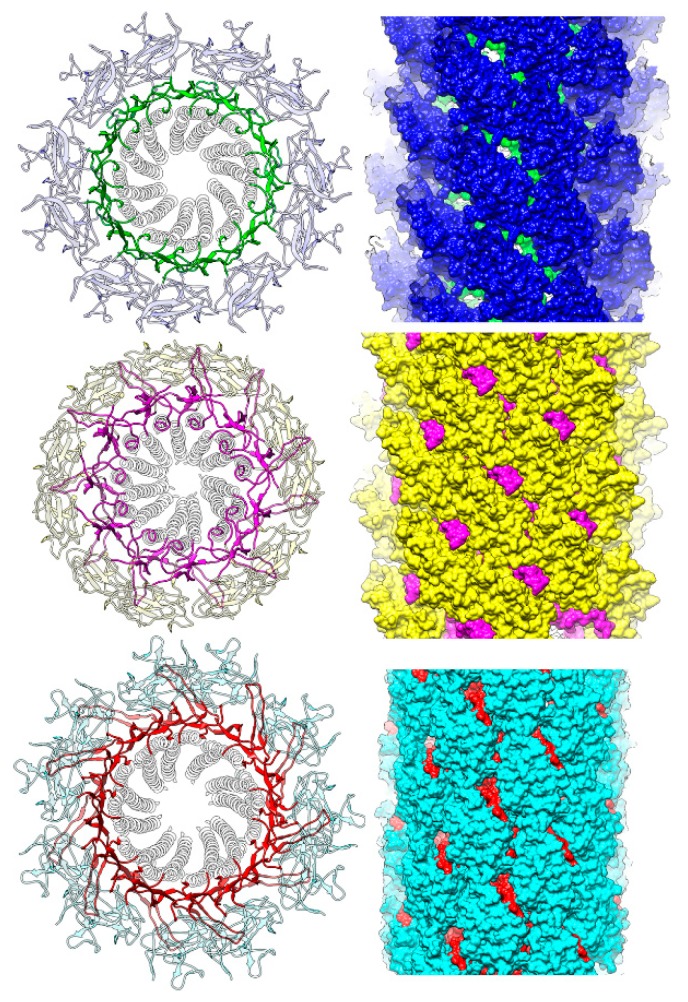
Structural comparison of *Salmonella* hook and rod and *Campylobacter* hook. The Cα ribbon models of *Salmonella* hook at the top, the rod in the middle, and *Campylobacter* hook at the bottom. The models are presented in the end-on and side views in the left and right panels, respectively, with only D0-Dc and D1 domains being shown for both hook structures. The Dc domains are highlighted in the left panels by different colors, and the D1 domains are also colored in the right panels. The gaps between the D1 domains are clear in the *Salmonella* hook structure at the top right, but the gaps are filled by the tip of the β-hairpin of domain Dc in the *Salmonella* rod and *Campylobacter* hook structures, further stabilizing and rigidifying their tubular structures.

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
