# Peer review of "Structure of Salmonella Flagellar Hook Reveals Intermolecular Domain Interactions for the Universal Joint Function"

_biomolecules, 2019, doi:10.3390/biom9090462_

Round 1
Reviewer 1 Report
This is a valuable study that shows the structure of the Salmonella hook at a resolution of 4.1 Å achieved by electron cryomicroscopy. The model obtained revealed the entire structure of FlgE building block of the flagellar hook and sheds information on the mechanisms that give the hook its characteristics.
I have minor comments that could improve the reading of this study.
Line
26. Change to, “at high enough resolution to allow building an atomic model of entire FlgE”
57. Insert, “direction causes a polymorphic transition”
58. Insert and delete, “...to fall apart and to cause cell tumbling”
88. Change, “FlgE subunits along the protofilament of the hook is are ...”
262. Insert, “important for flexible bending of the hook and for its universal joint function..”
282. Change, “structural stability that are is found between different domains.”
282. Insert, “Residues 38-40 in the..”
317. Insert, “central region of the FlgE sequence...”
318. Rearrange, “...and D2 which is responsible for the smaller diameter of the rod is missing.”
320. Should read, “..FlgG has 18 extra residues in the domain...”
322. Same issue, “..FlgG with 17 extra residues...”
327. Insert, “...FlgE that we report...”
362. Delete, “…D0-Dc are between these three molecules are...”
377. Delete, “...are shown for the both hook structures..”
389. Delete, “… spread apart and could not at all form a bundle...”
Author Response
Thank you for your positive comments on our manuscript and also for the list of comments for its improvement. We have incorporated your suggestions for most of the places.
For detailed response, see attachment.

Reviewer 2 Report
The bacterial flagellum is a nanoscale propeller used by bacteria to swim. One of the core components is the hook, which functions as a universal joint between the motor and the propeller, enabling bacteria with many flagella with different orientations to reorient and combine their rotating flagella together into a coherent bundle. This paper aims to better understand how the Salmonella hook works by determining its near-atomic resolution structure. The paper is an interesting contribution to the field. The authors report the complete structure of the Salmonella hook and use this, together with the previously-published structure of the Campylobacter hook by the group of Fadel Samatey to model the structure of the Salmonella distal rod. These three structures are then compared to gain insights into the function of the Dc domains (effectively, to stiffen the structure) and their contributions to different structure’s mechanical properties. I have no major concerns about the study. Minor points: Writing throughout is difficult to read, and many sentences are overly-long, especially in the Abstract. The text would benefit from a thorough edit for overly-long sentences. There is incorrect use of articles in many places, especially the inappropriate use of “the” throughout which would also benefit from a thorough edit (see, e.g., https://www.grammarly.com/blog/articles/) Line 89: use of the word “designed” is entirely inappropriate. The sentence would better read “The axial packing interactions of the FlgE subunits along the protofilament of the hool have evolved in such a way for the protofilament to be compressible […]”. Line ~93: recommend clarifying for the reader that D1 from different proteins may not be homologous. For example, although D0 from FlgE and FliC are homologous, D1 from FlgE and FliC are not homologous. This can be confusing for newcomers. Line ~93: recommend clarifying for the reader that D0 and D1 are not contiguous domains, but rather are composed of two separate stretches of amino acids from the two ends of the polypeptide. Line 130: “Observed” should read “imaged”. Line 130: what software used for data acquisition? Line 139: Helpful to spell out to the reader total cumulative electrons per square Angstrom. Line 194: “A few”: quantify. Line 255: I don’t agree that D2 won’t contribute to the (dimensionless) “structural stability of the hook”. Either qualify, specifically, what you mean by “structural stability of the hook”, or remove. Line 260: any hint of water molecule densities in the cryoEM map? Line 303: Confusing that red represents non-idential segments. Can this be changed? Line 322: Sentence is confusing: make it unambiguous that the “extra” 17 residues in Campylobacter FlgE are “extra” relative to Salmonella FlgE. i.e., Make it crystal clear that the Dc from Campylobacter FlgE is likely similar to Dc from Salmonella FlgG. Readers may appreciate some speculation at the end of the Discussion as to why the Campylobacter and Salmonella hooks likely have different stiffnesses? Is there something about their environment that would select for these differences? Figures would benefit from panel labels (e.g., “a”, “b”,…). Figure 1: difficult to see numbering for D0, can this be improved?Author Response
Thank you for your positive comments on our manuscript and also for the list of comments for its improvement. We have incorporated your suggestions for most of the places, replaced Fig. 1 with subunit labels in white in the upper right panel for clarity, and added a sentence in the last part of Discussion regarding a possible reason for the different stiffness of the hooks of Campylobacter and Salmonella.
For detailed response, see attachment.
